# Dispersal distances and migration rates at the arctic treeline in Siberia – a genetic and simulation based study

**Authors:** Kruse, Stefan[*,1,2], Gerdes, Alexander[1], Kath, Nadja J.[1,2], Epp, Laura S.[1], Stoof-Leichsenring, Kathleen R.[1], Pestryakova, Luidmila A.[3], Herzschuh, Ulrike[1,2,4]

[1]Polar Terrestrial Environmental Systems Research Group, Alfred Wegener Institute Helmholtz Centre for Polar and Marine Research, 14473 Potsdam, Germany
[2]Institute of Biology and Biochemistry, University of Potsdam, 14476 Potsdam, Germany
[3]Institute of Natural Sciences, North-Eastern Federal University of Yakutsk, 677000 Yakutsk, Russia
[4]Institute of Earth and Environmental Science, University of Potsdam, 14476 Potsdam, Germany

*Correspondence to*: Stefan Kruse (stefan.kruse@awi.de)

**Abstract.** A strong temperature increase in the Arctic is expected to lead to latitudinal treeline shift. This tundra-taiga turnover would cause a positive vegetation-climate feedback due to albedo decrease. However, reliable estimates of tree migration rates are currently lacking due to the complex processes involved in forest establishment, which depend strongly on seed dispersal. We aim to fill this gap using LAVESI, an individual-based and spatially explicit *Larix* vegetation simulator. LAVESI was designed to simulate plots within homogeneous forests. Here, we improve the implementation of the seed dispersal function via field-based investigations. We inferred the effective seed dispersal distances of a typical open forest stand on the southern Taymyr Peninsula (north-central Siberia) from genetic parentage analysis using eight nuclear microsatellite markers.

The parentage analysis gives effective seed dispersal distances (median ~10 m) close to the seed parents. A comparison between simulated and observed effective seed dispersal distances reveals an overestimation of recruits close to the releasing tree and a shorter dispersal distance generally. We thus adapted our model and used the newly parameterised version to simulate south-to-north transects: a slow-moving treeline front was revealed. The colonisation of the tundra areas was assisted by occasional long-distance seed dispersal events beyond the treeline area. The treeline (~1 tree ha$^{-1}$) advanced by ~1.6 m yr$^{-1}$, whereas the forest line (~100 trees ha$^{-1}$) advanced by only ~0.6 m yr$^{-1}$.

We conclude that the treeline in north-central Siberia currently lags behind the current strong warming and will continue to lag in the near future.

## 1 Introduction

Changing climate forces species worldwide to migrate (Arctic Climate Impact Assessment, 2004; IPCC, 2013). This is exceptionally challenging for sessile organisms such as plants as they may strongly lag behind their moving climate envelope (Harsch et al., 2009; Loarie et al., 2009; Moran and Clark, 2012). Warming is particularly pronounced in the Arctic where the tundra-taiga ecotone demarks the transition from forest stands to treeless areas and which is expected to move northwards (Harsch et al., 2009; Holtmeier and Broll, 2005). Such tree range expansion is of major interest because the establishment of forest in the dwarf-shrub tundra would reduce the surface albedo and promote a positive feedback to global temperature (Bonan, 2008).

Trees migrate via seed dispersal and face several ecological barriers (Svenning et al., 2014): first, viable seeds need to be produced, second, these need to be dispersed and, third, seeds need to germinate and survive to grow to new individuals. This process, called 'effective seed dispersal' (Connell, 1971; Janzen, 1970), determines the speed and spatial pattern of a species' response to climate change. For example, closely dispersed seeds and a long generation time result in a slow moving front, while a patchy pattern will form from many long-distance seed dispersal events (Clark, 1998; Nathan and Muller-Landau, 2000; Ritchie and MacDonald, 1986). Migration can speed up if there are relict trees from an earlier wider extent of forest which have survived in refugia ahead of the recent treeline (Stewart and Lister, 2001; Väliranta et al., 2011).

To project future species ranges, the potential migration rate under global warming is estimated via simulation studies (Kaplan and New, 2006; Roberts and Hamann, 2016; Snell and Cowling, 2015). However, these simulations depend strongly on the dispersal configuration of the model (Bhagwat and Willis, 2008; McLachlan et al., 2005; Stewart et al., 2010; Willis and Van Andel, 2004). Most empirical attempts to estimate historical migration rates are based on records of fossil pollen and macrofossils in sediment cores as indicators of species presence (MacDonald et al., 2008; Pisaric et al., 2001), but the interpretation is compromised because of a lack of knowledge about glacial refugia, particularly small 'cryptic' refugia that can be easily overlooked in the fossil record (e.g. Petit et al., 2008). Therefore, more reliable estimates of dispersal distances of tree taxa are needed in order to predict the treeline response under high-latitude warming (Snell, 2014; Snell and Cowling, 2015).

Understanding treeline changes on the southern Taymyr Peninsula is of particular relevance as the area is characterised by a strong warming trend (IPCC, 2013). It represents an ideal study area because the treeline is formed of monospecific tree stands of *Larix* Mill. Taxa and was thus the focus of several treeline studies (IPCC, 2013; Naurzbaev et al., 2002; Sidorova et al., 2010). The response to warming seems to differ with time-scale: while millennial-scale warming during the mid-Holocene is reflected by a treeline location 200 km further north on the Taymyr Peninsula (Andreev et al., 2002; Klemm et al., 2016; MacDonald et al., 2008), the decadal-scale ongoing warming generates no response (Niemeyer et al., 2015; Wieczorek et al., 2017), possibly because of low seed availability.

To study the responses and migration dynamics of treeline tree stands under climate change, LAVESI, an individual-based and spatially explicit simulation model for *Larix* (Kruse et al., 2016; Wieczorek et al., 2017), was developed. In comparison

to other dynamic vegetation models, it handles each individual larch tree beginning from a seed to an established seedling until becoming a mature tree and producing seeds itself and thus starting a new generation. This model includes wind-dependent seed dispersal and density-dependent growth and mortality processes. The representation of the full life cycle allows in-detail simulation experiments to unravel the influences of previously overlooked feedbacks (further details in Kruse et al., 2016; Wieczorek et al., 2017). However, the seed dispersal component had not been validated by observations. Traditional methods to track seed dispersal distances include seed traps and seed-bank analyses (Brown et al., 1988; Greene et al., 2004; Stoehr, 2000), which are time consuming and prone to underestimate distances (Ashley, 2010; Pairon et al., 2006). Fortunately, genetic analyses provide an alternative modern approach. Repetitive sequences in the nuclear genome (short sequence repeats, SSR, or microsatellites) are sufficiently variable genetic markers to resolve parentage (Ashley, 2010; Hartl and Clark, 2007; Schlötterer, 2000). Using such an approach, the dispersal of pollen and seeds in a landscape can be tracked and effective seed dispersal distances can be inferred (e.g. Pairon et al., 2006; Piotti et al., 2009; Pluess, 2011; Steinitz et al., 2011). For example, microsatellite studies have helped to elucidate the recruitment source of spruce juveniles and the dispersal patterns at an elevational treeline that recently shifted upwards (Piotti et al., 2009). Furthermore, a range expansion of larch following a glacier retreat could be tracked without a decrease in genetic diversity (Pluess, 2011). Genetic analyses can thus be used to provide a more realistic implementation of seed dispersal in simulation models.

With this study, we aim at improving seed dispersal and establishment processes in the simulation model LAVESI to make it applicable for simulating treeline migration rates. Therefore, we undertook a genetic parentage analysis of a treeline stand on the southern Taymyr Peninsula by applying an assay of eight nuclear microsatellites to get a reliable estimate of the effective seed dispersal distance (1). This information was used to improve the individual-based model LAVESI (2), which we then ran to simulate treeline advances into the tundra and estimate migration rates (3).

## 2 Methods 2.1 Sample collection

Needle samples from larch individuals (*Larix gmelinii* (Rupr.) Rupr) were collected from a tree stand during fieldwork in the summer of 2013 on the southern Taymyr Peninsula, Krasnoyarsk Region, in northern-central Siberia (plot name: TY04VI; 72.409 °N and 105.448 °E; Fig. 1). The open canopy forest stand with ~300 trees ha$^{-1}$ belongs to the forest tundra and has shown enhanced recent recruitment (site code FTe_1, Wieczorek et al., 2017). We sampled all individuals >0.4 m in height in a 20 x 20 m area as well as all trees >2 m high or bearing cones from the surrounding 100 x 100 m area (Fig. 3). Additionally, in the central 12 x 12 m area individuals <0.4 m were collected. Larch individuals from the 20 x 20 m plot were accurately mapped with a tape measure, while a standard GPS device (Garmin) was used to map the individuals in the 100 x 100 m area. We recorded the height of each individual and collected short twigs with needles and dried them in the field on silica gel.

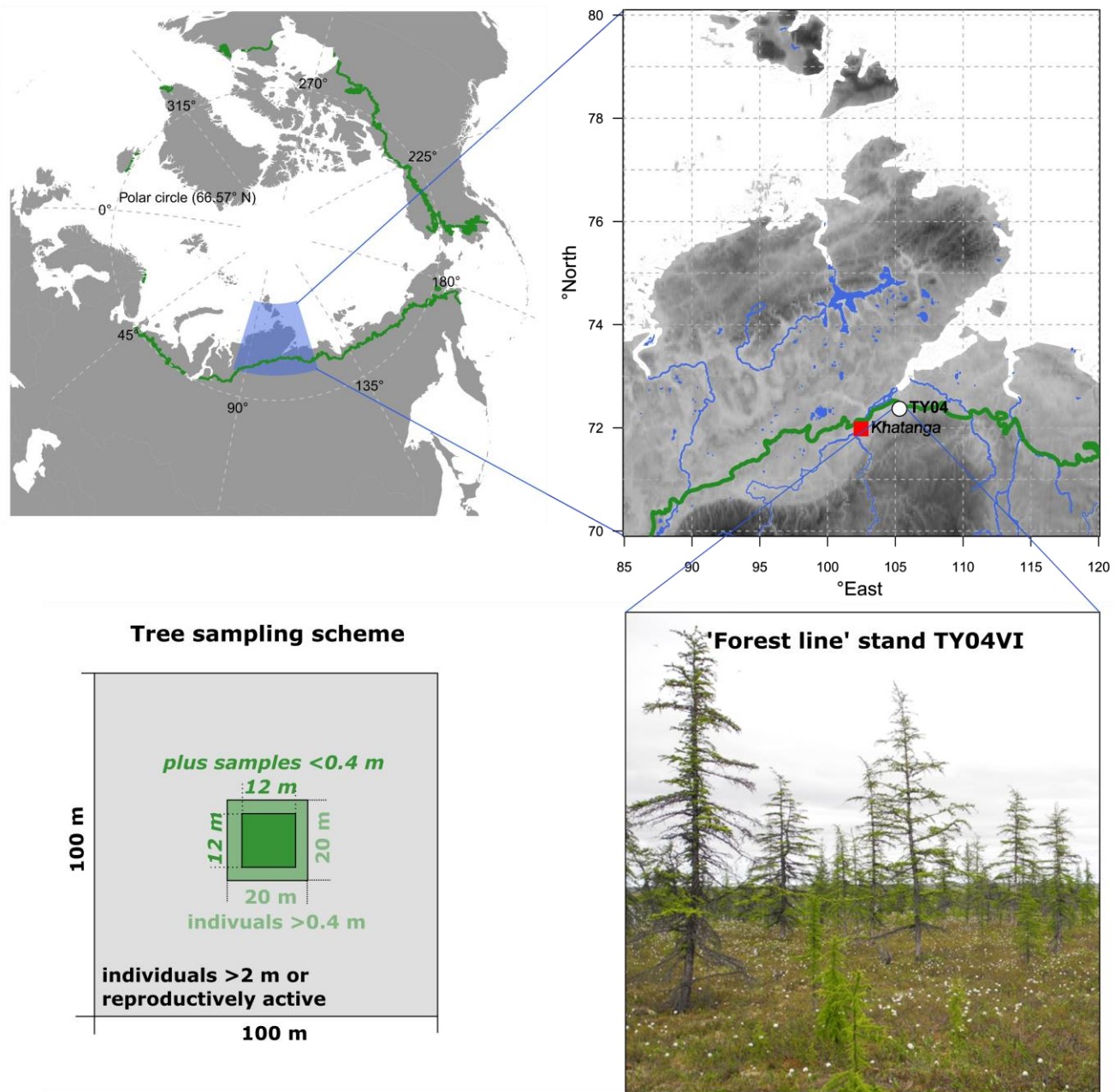

90

**Figure 1: Overview of the larch forests (*Larix gmelinii*) growing at study site TY04VI on the southern Taymyr Peninsula and the sampling scheme. The green circumarctic line on the maps marks the modern treeline (Walker et al., 2005). Topography in the enlarged area ranges between 1 and 2521 m, (WorldClim1.4 Hijmans et al., 2005). Rivers and lakes are given in blue colours (GSSHS updated Version 2.2.2 01.01.2013 first published by Wessel and Smith, 1996). Photo: Stefan Kruse, 16.07.2013**

95

## 2.2 Genotyping of individuals

Genomic DNA was extracted from 50–100 mg of dried needles after shock freezing in liquid nitrogen and grinding in a FastPrep® (MP BIOMEDICALS) with steel beads using the silica-column based extraction kit Invisorb® Spin Plant Mini Kit (STRATEC MOLECULAR). Subsequently, three multiplex PCR reactions (10 µl) were set up, including all eight primer pairs used in this study [bcLK211, bcLK253, Ld101, bcLK056, bcLK228, bcLK263 and bcLK189 (Isoda & Watanabe 2006) and Ld101, Ld42 and Ld56 (Wagner et al., 2012), further details in Supplement 1 Table S1] using the Multiplex PCR Master Kit (QIAGEN). Fragment length determination was done by SOURCEBIOSCIENCES (Oxford, UK). Allele sizes were scored in Geneious (version 7.1.5, BIOMATTERS LTD.) using the Microsatellite plugin (version 1.4.0). Raw allelic data were imported into R version 3.2.2 (R Core Team, 2015) and peaks binned to step sizes of two basepairs. The dataset was converted to the 'genind'-format using the package 'adegenet' (Version 1.4-2, Jombart, 2008; Jombart and Ahmed, 2011) for subsequent analyses. Details on the microsatellite primer selection, PCR-protocol and subsequent data analysis including binning of allele frequent lengths were described in Kruse *et al.* (Kruse et al., 2018a) and available online at https://doi.pangaea.de/10.1594/PANGAEA.885765.

## 2.3 Parentage analyses based on microsatellite data

The effective seed dispersal function was estimated from the results of a parentage analysis. We used eight highly diverse nuclear loci; five of which were sufficiently informative to distinguish all individuals, as is mandatory for parental assignment studies (Figure S2 in Supplement 1). We determined parents from allele frequency data with a likelihood-based approach implemented in CERVUS version 3.0.7 (Kalinowski et al., 2007). During the analyses, we allowed for 1% of errors in genotyping and a minimum of seven loci typed in the final analysis. All individuals (612 in total) were analysed and we searched for parents of recruits (height <2 m) from among all potential tree individuals (height >0.4 m). Following the program documentation we simulated in CERVUS the heritage for 10,000 seeds with a chance of 10% of a parent sampled and 1% error (Marshall et al., 1998; Slate et al., 2000) to determine thresholds for the 'log of the overall likelihood ratio' (LOD) scores in this analysis. Only those with positive assignments and a high confidence level exceeding 95% were retained for further analyses (Fig. 3b). However, we kept assignments where the parent-pair fell below the high confidence LOD threshold when one of the two parents could be assigned with high confidence. Assignments to identical genotypes were excluded from subsequent analyses (Fig. 3a). The goodness of the assignment was calculated as the mean of the exclusion probability over all tested offspring (which is one minus the non-exclusion probability calculated in CERVUS). Finally, we distinguished the more distant parent as the pollen donator (father) and the other as the seed source (mother) and all single parent cases as seed dispersal events following Dow and Ashley (1996).

## 2.4 Simulation study

### 2.4.1 The individual-based and spatially explicit *Larix* model LAVESI

We ran simulations with the individual-based and spatially explicit model LAVESI (Kruse et al., 2016; Wieczorek et al., 2017). This model simulates the life-cycle of larch individuals from seeds to mature trees and was parameterised for *Larix gmelinii*. On a simulation area of user specified size, individuals grow where seeds settle and germinate, and competition among individuals is handled by a fine sub-grid of cells with an area of 20 x 20 cm. The model was established to improve our understanding of past and future treeline displacements under changing climate. The relevant processes (growth, seed production and dispersal, establishment and mortality) are incorporated as submodules which are parameterised on the basis of field evidence (Wieczorek et al., 2017), complemented with data from literature. Seed dispersal into the environment is estimated by a Gaussian and fat-tailed dispersal function. With this and the detailed representation of competition the model realistically simulates, similar to Janzen's (1970) and Connell's findings (1971), that recruits have the highest chance to survive at intermediate distances to the producing tree, not directly at it. Fine-tuning the model parameters of involved processes, which includes the impact strength that competition has on smaller trees, allows adapting the effective seed dispersal distance. Simulation runs proceed in yearly time steps and are forced by monthly temperature and precipitation time series.

The original model of Kruse *et al*. (2016) was updated with the following processes (details in Supplement 2): (i) seed dispersal distances now depend on species-specific traits (tree height, seed properties) and wind speed and direction (Kruse et al., 2018b), (ii) the tree diameter growth function is newly calibrated to the climate forcing (Epp et al., 2018), and (iii) the active-layer thaw depth directly influences the tree's growth that is used to estimate it's seed production and mortality. Following these updates, the parameter settings of the original model were revised to simulate stands comparable to the site TY04VI (Kruse et al., 2018b).

### 2.4.2 Tuning the dispersal process in the model

We performed model runs to simulate larch stands in 100 x 100 m areas with closed boundaries. This means that seeds which leave the area on one side enter the field from the other side. To tune the model's processes in order to capture the observed effective seed dispersal distribution, we tested several combinations of model parameters and introduced new variables into formulae used in the program code of the model (listed in Table 1, details in Supplement 2 and in Table S5). Each simulation begins with a 2000 year-long spin-up phase, followed by an 80-year experimental phase (AD 1934–2013).

Climate forcing data was derived from the CRU TS 3.22 gridded data (0.5° resolution, Harris et al., 2014). We calculated the distance-to-centre weighted mean of all monthly temperature and precipitation values of the weather data of the TY04VI grid cell and its eight surrounding grid cells. May to August wind data were extracted from the ERA-Interim reanalysis data (AD 1979–2012; Bromwich et al., 2016). During the spin-up phase, weather data for each year was randomly sampled from the years AD 1939–2008 to allow the tree stand to reach a quasi-stable state. The period excludes a 5 year-long margin at the

beginning and end of the climate observations available from the Khatanga station (see Kruse et al., 2016 for further information). For years not covered by wind data, the model randomly selected a year from the available wind data series. Simulations were run for 10 repeats and the outcomes of all individuals present in the final simulation year of AD 2013 were recorded.

The distance from the seed source tree for each established individual, i.e. the effective seed dispersal, was inferred from the simulation results. We resampled these simulated distances to consider the same frequency of observed parenthoods in the central 20 x 20 m as in the surrounding 100 x 100 m area (sampling scheme details in section 2.1 sample collection). We included only simulations which had at least 10 individuals present in the central 20 x 20 m area in further analyses. Distances were binned to classes of one metre steps. We evaluated the simulation results by calculating the Pearson's product moment correlation coefficient between simulated and observed dispersal distances. Furthermore, we reconstructed the proportion of on–site reproductive success in the final year as the ratio between on-site recruitment and all recruits. The differences between simulated and observed on-site recruitment ratios were tested by one-sided Student's $t$-tests.

### 2.4.3 Simulation experiments to depict migration rate

The best model that resulted from the tuning process was used to simulate larch migration in a hypothetical area of 1000 m (east-west) by 5000 m (north-south). Tree growth was initialised during the first 100 years by introducing 20,000 seeds $yr^{-1}$ into the southernmost 100-m area. This setting was run under two contrasting climate scenarios: first, with homogeneous temperature and precipitation forcing named 'EvenClim', and second, with linearly decreasing temperatures from south to north – mimicking the real south-to-north climate gradient – named 'GradClim'. The gradients of mean annual temperature and annual precipitation are described by $-6.24*10^{-6}$ °C $m^{-1}$ and $+3.26*10^{-6}$ mm (year m)$^{-1}$, respectively, in a northward direction starting at TY04VI as inferred from an analysis of globally interpolated monthly climate data for 1960–1990 (Hijmans *et al.* 2005). Simulations under both scenarios were repeated 10 times and run for 2000 years using climate series from random years out of the available period of AD 1939–2008. Data for each established tree individual were collected every 10 years of the simulation.

We analysed stand densities for the entire simulated area. The number of trees ($>2$ m) was calculated from 100 x 100 m plots (=1 ha) by iteration over the entire area in 50 m steps (x and y). To reduce the errors introduced by the strict boundary conditions at the edges of the hypothetical area, the outer 100 m borders were excluded. The advance of larch stands into tundra was estimated from forest density by mean number of trees on east–west transects. We defined two relevant thresholds of tree densities (see also Fig. 2): (i) the 'treeline' when the mean tree density fell below 1 tree $ha^{-1}$ and (ii) the 'forest line' when mean tree density fell below 100 trees $ha^{-1}$. These thresholds are in accordance with observations in treeline areas in Siberia (Kharuk et al., 2006; Montesano et al., 2016; Wieczorek et al., 2017). The migration rates of these thresholds in metres advanced per year were calculated as the slope of a linear model describing the position of the 'treeline' or 'forest line' as a function of simulation time. The rates were tested with a two sample $t$-test for significant differences. Analyses were performed in R version 3.2.2 (R Core Team, 2015).

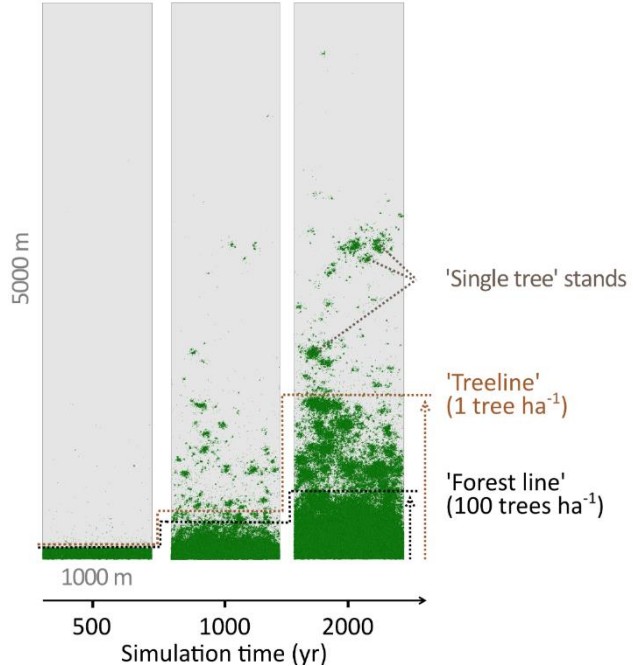


**Figure 2: Simulated treeline advances along a hypothetical south-north transect across the modern 'forest line'. Simulations were initialised with seeds in the southernmost 100 m area and run for 2000 years. In the beginning, trees established beyond the 'treeline' (mean density falling below 1 tree ha$^{-1}$) and formed 'single tree' stands in the tundra, which then acted as nuclei for further range expansion, so that in following years the 'treeline' and the 'forest line' (mean density falling below 100 trees ha$^{-1}$) could advance**

**northwards forming open forest**

## 3 Results

### 3.1 Effective seed dispersal distances as inferred from genetic parentage analysis

In total, 346 trees, 118 saplings and 148 seedlings were genotyped (Fig. 3). The mean distance between pairs of larch individuals was ~38 m (maximum: ~137 m). The mean tree height was 4.4 m (maximum: 9 m).

The eight chosen microsatellite loci were highly polymorphic and varied from 11 to 41 different alleles with only 0.49% missing alleles in total (Figure S1, Figure S4 in Supplement 1). The information content reached a plateau at four loci which could separate 597±2 individuals (>99%) and the power increased slightly towards 600±1 separated individuals with seven

loci (Figure S2 in Supplement 1). Accordingly, we included all eight loci in the subsequent analyses to separate all individuals. In total, 601 sampled trees could be distinguished and 22 individuals were identified as 10 clonal groups, of which 11 were subsequently excluded from further analyses (Fig. 3a, Supplement S1). The maximum distance between two individuals within these groups was 30 m but mostly <5 m (Fig. 3a).

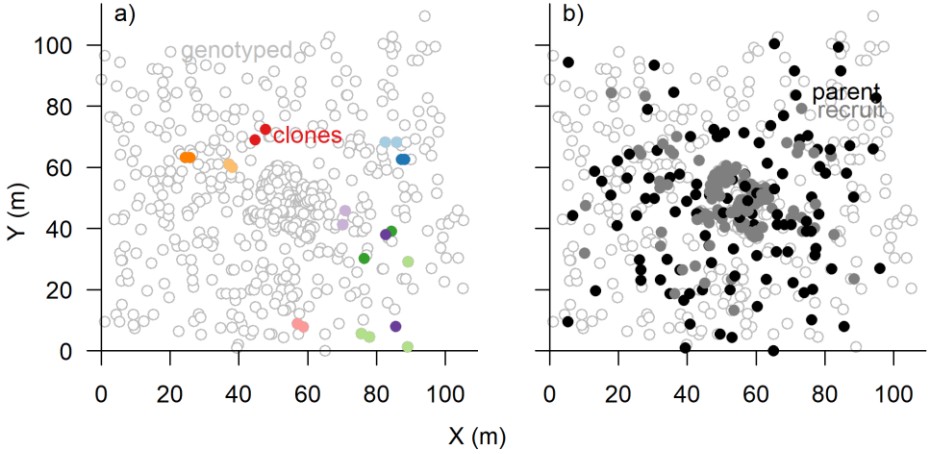

**Figure 3: Map of genotyped *Larix gmelinii* individuals at site TY04VI, individuals sharing the same genotype (clones) are marked by filled points of the same colour in a) and local recruits are marked by filled bright circles whereas parents by filled black circles in b)**

In the parenthood analyses, we aimed to find the parents for 266 individuals (<2 m) from among the remaining 464 individuals

(>0.4 m). The exclusion probability for one parent was ~99.9910±0.0172% and for a pair of parents 100% in the used assay. A single parent or both parents were assigned for 151 individuals with a high confidence (>95%; LOD threshold for the parent pair of 15.13 and for a single parent 5.76, examples in Fig. 4). This is ~53% on-site recruitment in respect to all tested offspring. Among these, in 49 cases we found both parents (18.4%) and for 92 only a single parent (34.6%) was assigned. One of the largest trees (H=7 m) was assigned to 8% of the recruits (Fig. 4). Trees with many assignments are generally larger than those

with few (Figure S5 in Supplement 1). Mostly parental trees exceeded their offspring's height, but in 3.7% cases recruits were higher than their assigned parents.

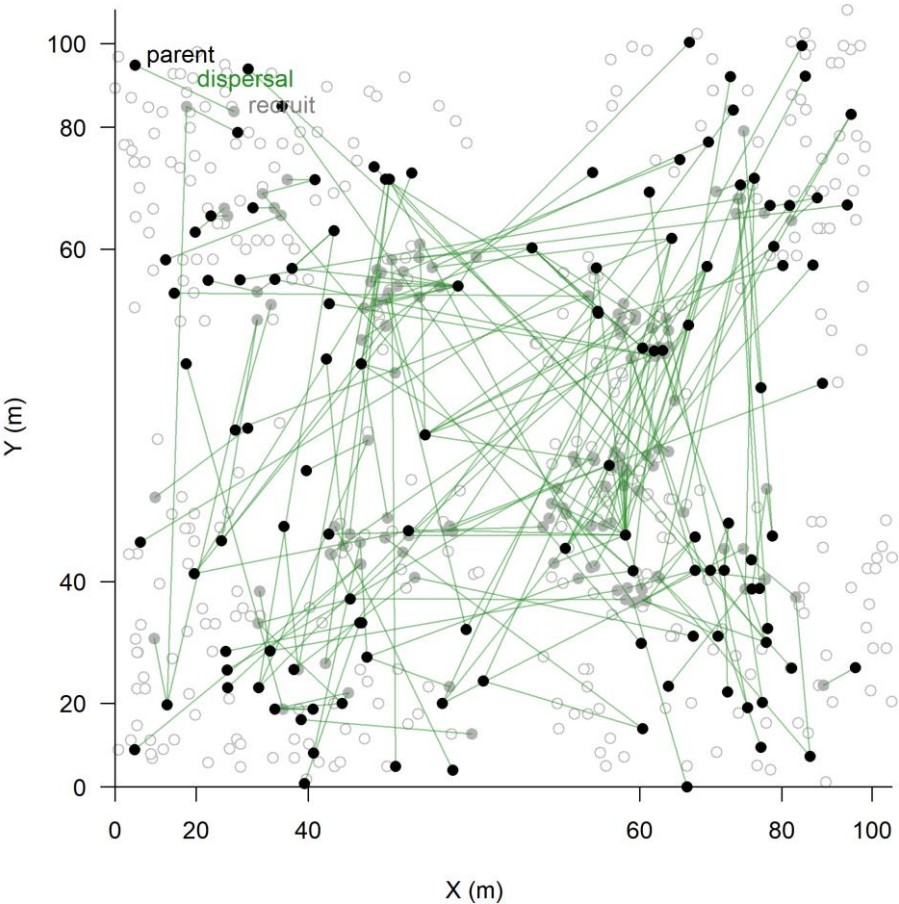

**Figure 4: Map of assigned relationships between parents (filled black circles) and their offspring (filled bright circles) of *Larix gmelinii* individuals. Additional genotyped individuals are given as open circles. Note the non-linear scale of the coordinates**

We identified 150 effective seed dispersal distances when assuming that the closest parent is the seed source when two parents were assigned or only a single parent was identified. The observed mean distance of effective seed dispersal is ~15.0 m (median

of ~9.8 m), with a minimum of 0.8 m and a maximum of 56.1 m (Fig. 5).

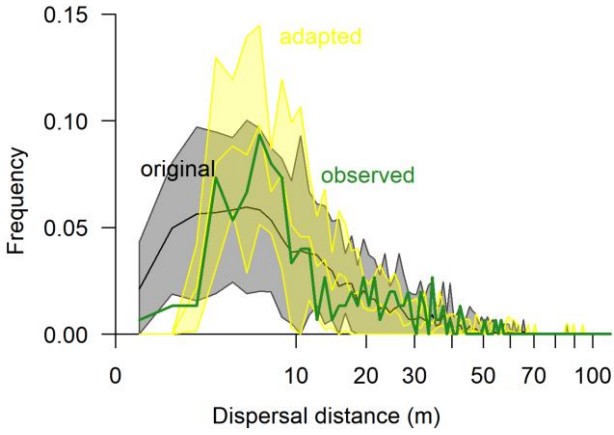

**Figure 5: Seed dispersal of *Larix gmelnii* as simulated with the original simulation model LAVESI (open circles, grey band shows the 95% confidence interval) and the adapted model version (filled bright circles; yellow band shows the 95% confidence interval) compared to observed effective seed dispersal (filled dark circles). Dispersal distances were binned to one metre classes**

## 3.2 The adapted effective seed dispersal in LAVESI

In the original model (Kruse et al., 2016; Wieczorek et al., 2017), effective seed dispersal distances follow a right-skewed distribution (Fig. 5). Mean dispersal for this model is 13.5 m (median 9.98 m), with a minimum of 0.6 m and a maximum of 60.3 m. In general, it captures the observed data, but it over-emphasises the recruits close to the mother tree at distances of 1–3 m. Furthermore, the simulated maximum probability peak between distances of ~4–7 m is roughly 3 m closer to the seed source than observed. Also, the observed tail approached zero probability faster at the far distances compared to the observed effective distances. These three deviations were used to guide the model adaptation.

The best-fitting model has effective dispersal distances that match well to the observed distances (r=0.93) and has the smallest sum of residuals (0.0063) compared to the other parameter sets (0.0066–0.1623). It is driven with a combination of parameters which increase the dispersal (Sdist=1, $r_{GaussExpDisp}$=1.0 and $d_{GaussCentre}$=4.0) and seed production rate ($f_s$=11) compared to the original model (parameter set "I": Figure S7, Figure S8, Table S5 in Supplement 2). The resulting mean dispersal distance is 12.3 m (median 8.85 m) with a range of 2.7–71.1 m (Fig. 5).

The on-site recruitment ratio (~53%) was generally underestimated in the simulations (38.1–54.9%, Table S5). The parameter settings which produced similar ratios greatly overemphasised dispersal at short distances and, thus, the simulated seed dispersal pattern deviated strongly (correlation values of r=0.26–0.58, "tCJ"). The simulations with the best-fitting model overemphasise effective dispersal distances so that more recruits immigrate into the plot (~7%, Supplement 2).

### 3.3 Simulating migration dynamics in the taiga-tundra transition zone

Simulations were run for a hypothetical 5000 m long south-north transect, initialised by introducing seeds to the southern 100-
255 m wide area. In the homogeneous climate scenario, EvenClim, single trees spread up to ~3600 m during the 2000-year
simulation and a 'treeline' (mean density falling below 1 tree ha$^{-1}$) formed at ~2000 m. A 'forest line' (mean density falling
below 100 trees ha$^{-1}$) formed up to ~500 m further north than one forced by the climate gradient scenario ClimGrad (Fig. 7).
Migration was first accelerated by isolated colonisation events above the 'forest line', so that the 'treeline' moved northwards
by ~1.5 m yr$^{-1}$ into treeless areas, but decelerated after a peak between 500–1000 years (Fig. 6). The advance of the 'forest
260 line' on the other hand accelerated throughout the EvenClim simulation until becoming six times faster at the end of the
simulation (1500–2000 yrs) than at the beginning (0–1000 yrs). The migration rates of the 'forest line' were roughly half in
the GradClim scenario.

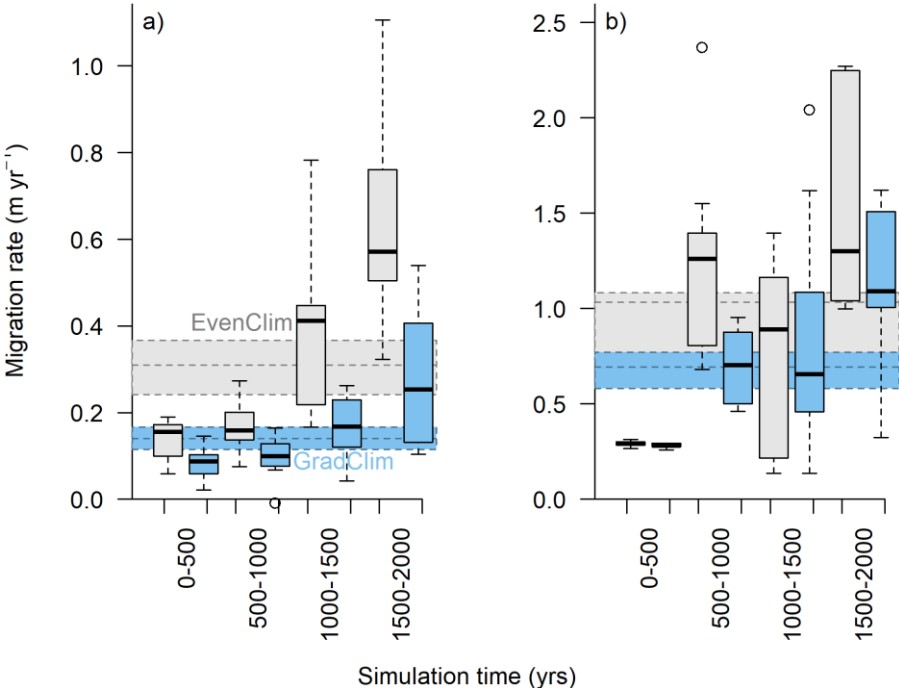

**Figure 6: Simulated migration rates of the 'forest line' (mean density falling below 100 trees ha$^{-1}$) and the 'treeline' (mean density**
265 **falling below 1 tree ha$^{-1}$) estimated from the best-fitting model. The simulations were forced by two contrasting climate scenarios,**
**either homogeneous temperature across the area (EvenClim: grey shading) or linearly decreasing temperature from south to north**
**(GradClim: darker blue shading)**

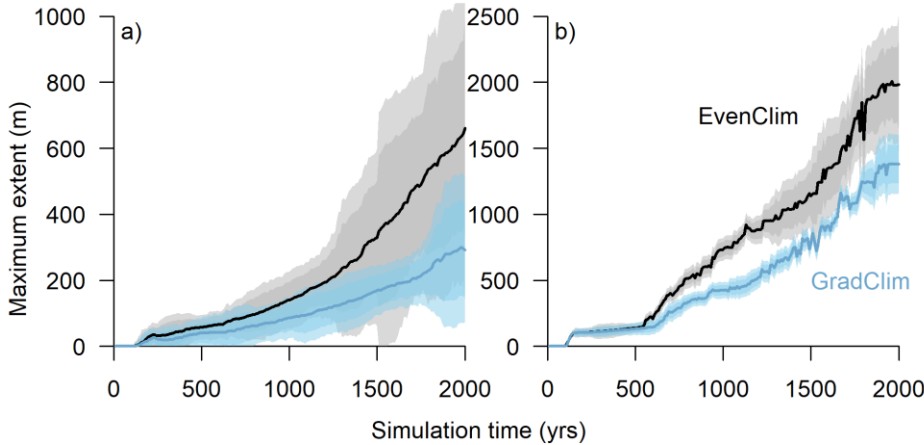

**Figure 7: Simulations along a hypothetical transect at the taiga-tundra transition reveal the northward advance of a) the 'forest line' (mean density falling below 100 trees ha⁻¹) and b) 'treeline' (mean density falling below 1 tree ha⁻¹). Simulations were forced by two contrasting temperature scenarios (homogeneous temperature EvenClim (grey shading) and northwards linearly decreasing temperature and precipitation GradClim (darker blue shading)). Shaded areas give the 99 and 90% confidence intervals around the mean value of 10 simulation repeats**

## 4 Discussion

### 4.1 Seed dispersal distances inferred from genetic heritage analysis

Our assay of eight highly polymorphic microsatellites distinguished all 601 genotyped individuals allowing us to infer the local recruitment pattern of a tree stand and, from that, the effective seed dispersal distances. In this analysis, we needed to exclude the observed clonal groups that are consequences of exceptional reproduction. We are confident that these are true observations of clones as we minimized the chance of full sibs share the same genotype by using highly polymorphic nuclear microsatellites that are not in linkage disequilibrium (Kruse et al., 2018). Nevertheless, we cannot rule out that selfing or back-crossing have occurred that could yield to offspring being genetically identical to one of the parents. If those modes of inheritance regularly occur and would have caused a misidentification of full siblings as clones, we would expect to observe an continuously increasing number of transitional states from identical genotypes (0 different alleles) to sharing 50% of their alleles (8 different alleles). However, it sharply drops from the identified clonal groups to a very low value and increases again beginning at 3 to 4 differences (Fig. S6 in Supplement 1). This gives us confidence to classify such identical individuals as clones. An explanation for these could be, that wind thrown trees can survive or in non-favorable conditions producing horizontal branches rather than upright stems forming krummholtz (own observations, Wieczorek et al., 2017). By producing adventitious roots from branches touching the ground (Kajimoto, 2010; Cooper, 1911) and subsequent separation of the main stem or horizontal branch two genetically identical individuals can be found if both parts survive.

We were able to identify at least one parent for a majority of the offspring in the parentage analysis (53%: 18.4% both parents and 34.6% one parent), even though only those cases with a high degree of confidence (>95%) were regarded and an area of

only 100 x 100 m was analysed. Unfortunately, the labour-intensive sample collection and genetic analyses restricted the analysis to a rather small area in comparison to the large area of the treeline transition zone. Assessing the parentage across a broader scale and for different positions in the treeline ecotone would further help to understand dispersal dynamics at the treeline but the additional knowledge gain does not scale with effort. The observed effective seed dispersal ranged between ~1 and 56 m (median: 10 m). This aligns with the short dispersal distances generally reported for larches. For example, it was found that most seeds (94%) fell within 18 m of the releasing trees in a study of *Larix laricina* in northern USA (Duncan, 1954). This result, however, is not directly comparable to effective seed dispersal, because all dispersed seeds are included in the estimation and not only those which germinated and established as a new individual. Effective seed dispersal distances of 2–48 m were found in dense forests of *Larix decidua* in the Swiss Alps using an approach similar to ours (Pluess, 2011). Higher effective seed dispersal distances have been observed, however, for other wind-dispersed tree taxa (e.g. 27–58 m for *Pinus pinaster*: González-Martínez et al., 2002; 39–833 m for *Picea* Piotti et al., 2009). One explanation for the observed differences might be the density of the tree stands because established trees reduce the wind speed. In our tree stand, which is comparatively denser than those studied above, shorter dispersal distances are more likely than in open areas (Antonovics and Levin, 2016). Furthermore, dispersal is dependent on the release height (Matlack, 1987), which in our stand, was rather low due to the shortness of the trees (mean: ~4.5 m, max: 9 m). Most cones occurred on branches at roughly half the tree's height (see Fig. 1), as is typical for open stands.

The observed amount of on-site recruitment is high compared to other studies (Piotti et al., 2009; Pluess, 2011), but lower than that found in orchards (Funda et al., 2008). We found both parents for one fifth of the offspring within the analysed area (compared to only 11.1% in Piotti et al., 2009). As a result, recruits effectively immigrated at a rate of 47% from the exterior of the analysed 100 x 100 m area, which is similar to a study of *Picea abies* in the Italian alps (43.3%: Piotti et al., 2009).

The parenthood in our investigated site was dominated by a few, relatively tall trees. For example, a 6 m-tall tree generated 8% (17 cases) of all identified recruits. This observation is reasonable, as the chance of producing viable offspring increases with age and size, and, once a tree is well established, its seeds are released into the local environment. Other studies make similar observations. For example, Dow & Ashley (1996) found that *Quercus* saplings often established close to the releasing tree and the majority of the offspring were assigned to four out of 62 mature trees. Schnabel *et al.* (1998) observed that three *Gleditsia* trees produced 58% of the offspring and Piotti *et al.* (2009) found that six local adults produced ~62% of *Picea* juveniles.

Overall, our results indicate that incorporating individual seed dispersal (such as implemented in LAVESI) rather than introducing a certain seed sum needs to be implemented in models to realistically model tree migration processes.

We were unfortunately unable to distinguish between fatherhood and motherhood using nuclear inherited markers (DeVerno et al., 1993; Szmidt et al., 1987). It is a valid criticism that we simply assume that single-parent assignments represent seed dispersal events (following Dow and Ashley, 1996; Moran and Clark, 2012; Piotti et al., 2009). In the extremely unlikely case that the more distant parent was instead the seed source (results not shown), the effective dispersal distance would increase to a median of ~23 m. This would lead to a further decrease in on-site recruitment which is already slightly underestimated.

Furthermore, our approach risks assigning missing parents to extra-site recruitment if the local parents have died, leading to an overemphasis of the fat tail. We consider this risk to be low in our analysed tree stand as we found only a few dead trees or saplings within the 100 x 100 m area, and they were already largely decayed (Wieczorek et al., 2017).

## 4.2 Genetic-model comparison and model adaptation

Using observations of parentage from a treeline stand of *Larix gmelinii* we improved the seed dispersal function in LAVESI so that it will better represent larch migration.

The original dispersal function modelled seed distribution using a simple Gaussian density function with a fat tail (Kruse et al., 2016), as is implemented in a number of models (e.g. Levin et al., 2003; Snell, 2014), but, in contrast to most other models, dispersal in LAVESI is related to wind speed and direction (Kruse et al., 2018b). The most realistic simulation results are

achieved via a combination of parameter adjustments, i.e. shifting the implemented Gaussian distribution term by 4 m away from the centre, increasing the factor scaling the distance by roughly six times the original value and increasing the influence of the Gaussian term twice (model "I" in Table S5 in Supplement 1). With these adjustments, the simulated effective seed dispersal distance aligns fairly well to the observed values. The new function slightly overestimates dispersal and therefore allows ~7% more recruits to immigrate into the plot. This discrepancy, however, might also be an artefact related to the

shortcomings of the genetic parentage analyses. Regardless, with the dispersal function parameterised to the observed effective seed dispersal, simulations are now more realistic than with the original version of the model (Kruse et al., 2016).

## 4.3 Treeline migration rates

We performed simulation experiments with the best-fitting model to estimate the potential migration rate of the treeline on the southern Taymyr Peninsula. Under the scenario of even temperature and precipitation (EvenClim), the northwards migration

rate of the 'forest line' is ~0.6 m year$^{-1}$ and the 'treeline' ~1.6 m year$^{-1}$. Under the more realistic climate gradient scenario of northwards decreasing temperature and slightly increasing precipitation, an even slower advancing 'treeline' and 'forest line' is implied. Overall, we find an astonishingly low migration rate, even though the best-fitting model slightly overestimates immigration at the stand level. Our simulations may yet be conservative as we cannot rule out that the dispersal function underestimates far distance dispersal at the same time as overestimating intermediate distance dispersal. Nevertheless, the slow

recruitment ahead of the 'treeline' is in accordance with field observations at northernmost single-tree stands in the tundra, which show creeping growth-forms (krumholtz) and no apparent recruitment (Wieczorek et al., 2017). Our estimated migration rate is quite slow compared to the observed spread of larch individuals into the tundra by 3–11 m year$^{-1}$, as mapped by Kharuk et al. (2006). However, the stand Kharuk et al. (2006) investigated is an exceptional open-forest island close to a river ahead of the modern 'forest line' where winds might be stronger leading to higher dispersal rates (Antonovics and Levin, 2016;

Duncan, 1954). Another field-based study reports a treeline expansion of 50 m year$^{-1}$ in arctic Alaska (Lloyd, 2005), whereas an elevational range shift for larch in the Polar Urals of 20–60 m during the last century is reported by Devi *et al*. (2008) and a general upward shift of 20–50 m between 1910 and 2000 of open forest in this mountainous area (Shiyatov et al., 2005;

Shiyatov and Mazepa, 2012). During the Holocene Thermal Maximum boreal forests expanded on the Taymyr Peninsula to their northernmost position during the Holocene, which was likely assisted by glacial refugial populations ahead of the treeline

(MacDonald et al., 2000, 2008). The treeline responded with a centennial lag to environmental improvement, for example solar insolation, and reached its maximum position at ~8000 to 4000 yr BP, and subsequently declined to reach its modern limits around 3000 yr BP (MacDonald et al., 2000). Recently, global warming is ameliorating conditions for *Larix* forests in Siberia and evidence can be found that treeline stands are starting to respond, but at a slower rate than one might expect given the strong increase in temperatures (Wieczorek et al., 2017; Harsch et al., 2009). A possible explanation for the slow advance may

be because we report the advance of a forest line rather than single trees. Furthermore, we analysed only one tree stand and effective dispersal rates will likely differ among sites depending on a variety of abiotic or biotic factors (Moran and Clark, 2012). The actual dispersal distance depends on stand density, amongst others, as more trees reduce the wind speed (Antonovics and Levin, 2016) and establishment will be affected by local density-dependent mortality due to seed predation close to the releasing tree (Janzen–Conell effect, Janzen (1970) and Conell (1971)). Furthermore, the probability of seeds

surviving and forming a seedbank and the survival rates of seedlings strongly determine the colonisation speed. This is linked to the availability of microsites where seedlings benefit from shelter, thus lowering their mortality rates (e.g. Resler et al., 2005; Maher et al., 2006; Germino et al., 2011). These effects are not explicitly simulated but implicitly taken account of by our model parameterisation (Kruse et al., 2016). Migration corridors along rivers are not taken into account but they likely assist colonisation in these landscapes because of deeper active-layer depths close to the rivers and also from downstream seed

dispersal (Neilson et al., 2005; Wieczorek et al., 2017). Nevertheless, the positive impact of an increased survivorship on migration rates can be observed in our migration simulation experiments.

The mortality rate ahead of the treeline is lower under homogeneous climate than in the linearly decreasing climate gradient scenario with the consequence that the migration enters the exponential phase earlier (Fig. 6 & 7). In addition, we based our model adaptations on an area that is only one hectare in size and with this we cannot directly assess the long-distance seed

dispersal to which to fit our implemented kernel. To account for these cases, we implemented a Gaussian dispersal kernel combined with an exponential shaped with a fat tail (Kruse et al., 2016). In this study, this allows numerous seeds to be dispersed to far distances and led to a higher immigration into the simulated forest plot than observed. In consequence, the simulated migration rate tends to be overestimated.

This comprehensive study from genetic analyses to a model application is a first attempt showing the importance of

undertaking these timely model parameterisation studies and should be enhanced by, for example, inferring the parentages for other positions in the treeline ecotone on the southern Taymyr Peninsula.

Our results show that small uncertainties in the implementation of dispersal in a model impact the timing and shape of the simulated tundra colonisation. This is in accordance with a simulated lag in vegetation response to climate change when seed dispersal in a global dynamic vegetation model is constrained rather than using the usual unlimited seed bank approach (Snell,

2014; Snell and Cowling, 2015). However, further processes on smaller scales can constrain the response of tree stands and should therefore not be neglected in simulation studies: an advancing front is shaped by short-distance dispersal and spatially-

explicit processes, such as competition between individuals. A simulation model with spatially-explicit seed dispersal combined with a representation of small-scale population processes helps to give realistic estimates on the migration rates. We have demonstrated that the LAVESI model allows a realistic implementation and parameterisation of dispersal processes.

In summary, our results suggest that the current climate change will lead to a lagged response by decades to centuries. In particular, the first step of migration will be slow, although the subsequent infilling could be rapid. It seems likely, therefore, that recent strong warming will cause a highly nonlinear response in forest and treeline advance.

## 5 Conclusions

We parameterised and applied the individual-based model and spatially explicit LAVESI to estimate migration rates of the
treeline and forest line advance under current climate conditions. First, we inferred the effective seed dispersal distance from a genetic parentage analysis based on nuclear microsatellites, and second, we improved the dispersal process of the model according to the observed dispersal pattern.

In our genetic analyses, we found a genetically diverse tree population at a location within the treeline close to the tundra in Siberia. The parentage analysis revealed that the majority of recruits (~60%) have a local origin. Knowing the positions of the
parent trees, we could estimate the effective seed dispersal distances between parent and offspring, which are mostly short (~10 m), although longer distances (up to ~60 m) are possible. Simulations with the adapted LAVESI model improved our knowledge about the likely treeline migration response. The rate is surprisingly slow: just a few metres northwards per year. To find out if the estimated slow migration is an outlier coming from overfitting to only one study site or the general response rate under current warming, further similar studies at other treeline positions would be necessary. The simulated migration
pattern also showed that occasional long-distance seed dispersal events far beyond the treeline area assisted the colonisation of the tundra. Our migration rate estimates are in the lower range of those observed and significantly slower than those inferred from palaeoecological studies or from simulated vegetation responses to climate change in dynamic global vegetation models. These findings indicate that the treeline in north-central Siberia will lag behind the recent strong warming (which is moving by ~1,000 m yr$^{-1}$, Loarie et al, 2009) but if isolated trees occasionally establish in the tundra, they could become nuclei for a
rapid colonisation of the tundra. Should this rapid colonisation occur, the albedo of these populated tundra areas will reduce and thus a positive feedback to climate warming will follow the lagged response of tundra-taiga transition. Such a scenario could be run in a large-scale simulation experiment using the improved version of the LAVESI model in an attempt to learn more about the impacts of such a vegetation-climate feedback in the upcoming decades.

## Code availability

The source code of the original model LAVESI is available at GitHub https://github.com/StefanKruse/LAVESI/releases/tag/v1.01, and stored in the zenodo database

http://doi.org/10.5281/zenodo.1155486. The updated version with a wind-dependent seed dispersal kernel named LAVESI-WIND is available in the first version 1.0 and accessible at GitHub at https://github.com/StefanKruse/LAVESI/tree/v1.0 and stored at http://doi.org/10.5281/zenodo.1165383.

**Sample availability**

Sampling locations, morphological data and microsatellite genotype data will be made publicly available at PANGAEA after acceptance of this manuscript.

**Author contribution**

SK and UH designed the study and SK performed the experiments. SK, LSE, KRS-L and LAP conducted fieldwork. SK
generated molecular analysis. SK, AG and NK implemented the model and performed model simulations. SK, LE and UH analysed the data. SK wrote a first version of the manuscript that all co-authors commented on.

**Competing interests**

The authors declare that they have no conflict of interest.

**Acknowledgements**

We thank our Russian colleagues from the joint Russian-German expedition 2013 for support in the field. Special thanks to Alexey Kolmogorov, Bastian Niemeyer and Xenia Schreiber for their help in sampling and tree measurements. We would like to thank Cathy Jenks for proofreading and improving the manuscript. Furthermore, we thank the Associate Editor Kirsten Thonicke and four anonymous reviewers for their comments improving earlier versions of the paper. The PhD position of Stefan Kruse was funded by the Initiating and Networking Fund of the Helmholtz Association and the ERC consolidator grant
Glacial*Legacy* of Ulrike Herzschuh (Grant No. 772852)..

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
