# Peer review of "S1 Genetic analyses"

_Biogeosciences, 2018_

## Referee Comment (RC1) · Anonymous Referee #1 · 19 Sep 2018

General comments:

The manuscript by Kruse et al. describes an empirical study of effective seed dispersal using molecular markers which is used to adapt and parametrize a simulation model on larch migration rates at the arctic treeline. The topic of this study is of general interest because estimates for the capability of species to shift their distribution ranges are important for assessing the impact of climate change on many ecosystems. Especially tree species are of interest since they are the foundation species of many ecosystems and exhibit life history traits, which make direct observations difficult. The study offers a nice example for a combined approach with empirical data and simulations

although its direct implications are somewhat limited due to the lack of replications and the rather small study site. However, the authors acknowledge these limitations in their manuscript and it will be nice to see some replicates in the future to judge the range of migration rates possible at different locations.

The manuscript is overall well written and clearly structured. The applied methods are well chosen and experiments and data analyses are described in sufficient detail. The results are discussed in a concise way using the available body of literature and conclusions are well founded.

In general the manuscript is of high quality and I have not found any major flaws. Please find below a few specific comments.

Specific comments:

Page 2 lines 53 ff: These sentences are a bit hard to follow. Also it does not immediately become clear that the authors refer only to the Taymyr peninsula. Since there is so much literature on this available, it should maybe added somewhere that this region is well studied, which is a further argument for chosing this region for the study.

Page 3, lines 70 ff: I suggest to elaborate a bit more on the specific aims of the study here at the end of the introduction. Some aspects have been mentioned in earlier paragraphs but rather indirectly and not specifically related to this study.

Page 9, lines 195 f: Is there an explanation how ramet pairs can occur 30 m apart in Larix? How about the chance for full sibs to have identical genotypes?

Page 15, line 320: When I read this, I asked myself if the model includes the case of established individuals ahead of the treeline, which are not able to reproduce, yet, because conditions do not allow this at the moment. When the conditions change, the treeline might progress quite rapidly at first and then slow down. Since I am not familiar with the model in detail, I cannot judge if this is a point worth discussing or a scenario worth simulating.

Page 15, lines 325 ff: The migration rates mentioned here, are they 20-60 m/20-50 m for the entire time period respectively or per year in these time periods? Is it possible to translate the elevational shift into a migration rate comparable to the model?

Page 15, line 330: Establishment will for sure be affected not only by density-dependent mortality but also by abiotic conditions and their stochasticity in this extreme region of the planet.

Technical comments:

Page 5, line 102: "inferred" Why are the microsatellite data described as inferred? To me they seem quite directly measured.

Figure 5: The x axis is quite cramped in this figure. Maybe it could be stretched out a bit?

---

## Referee Comment (RC2) · Anonymous Referee #2 · 26 Oct 2018

This paper uses the LAVESI model to explore the rate of northward Larix migration on the southern portion of the Taimyr Peninsula in Siberia. The study uses recent field data collected in the area to drive the simulation work. The focus of the model parameterization/calibration was to improve the Larix dispersal functionality using genetic parentage, and then apply the model to understand how the Larix dispersal characteristics will play out in given some climate scenarios. The site of this modelling work is of particular interest, as it is near the northernmost forest stands where paleoecological records have indicated past presence of trees north of where they are currently found. Given the unique nature of Larix forests (deciduous conifers linked to continuous permafrost distribution) and the fact that across their broad spatial domain (central and

eastern Siberia) this class of trees coincides with marked changes in climate, and the potential for changes in tree distribution to alter the dynamics of high latitude systems (through changes to albedo and permafrost), this study is of great interest. I think the paper is for the most part clear and well-structured. Perhaps there can be some modification of the Discussion based on my main critique.

Main critiques: 1. At the scale of this study, are south-north assumptions of tree migration robust? At fine scales, the migrations may occur according to the patterns of favorable microsites, and primarily be confined to corridors with favorable active layer dynamics, and direct insolation. Such landforms seem important for explaining the current pattern of trees. A study with such an individual model that doesn't account for the conditions that are associated with the germination, survival, and growth of those individuals that are being dispersed should probably discuss in some detail this issue of the micro-site constraints that may contribute to broad error bars associated with migration rates. In other words, would the velocity of migration change between different microsites? If so, what is the relative prevalence of such favorable sites across the landscape, and how are they connected to the seed sources? These may be questions for follow-up work, and may be beyond the scope of this study, but I think a section in the Discussion could serve as a link between this study and some potentially viable next steps (one that incorporates landforms and micro-sites details).

2. This area is the northernmost forest ecotone. Some discussion for why this may be the case (paleoecological history) could be interesting and help contextualize predictions of future treeline velocities.

3. In Figure 1, a simple schematic of the sampling design could be useful.

Other comments: In the abstract, might it be possible to replace some of the technical wording associate with the genetic analysis with other more recognizable terminology that would be more likely to be understood by most of the readers of this journal?

Figure 4. Nice figure that shows how a few individuals dominate the reproduction. I

think it needs to be graphically enhanced. Suggestion: Put an alpha (i.e., transparency) on the green lines, and put the dots on top.

Section 4.3 Line 336 - "However, further processes . . . and should therefore [NOT?] be neglected in simulation studies:..."
* * *

---

## Referee Comment (RC3) · Anonymous Referee #3 · 20 Nov 2018

This study presents seed dispersal improvements made to the LAVESI individual-based, spatially explicit vegetation simulator to investigate larch migration rates on the Taymyr Peninsula in Northern Siberia. The seed dispersal equation updates were based on a field-based study of genetic parentage of trees in the study area, which found that prior to improvement, the model tended to underestimate dispersal distances in general and overestimate the numbers of recruits close to the parent tree. The updated model was used to simulate south-to-north transects and the rate of tree line advance was found to differ from the rate of forest line advance by ∼1m per year. The study is well-structured and presented, and it addresses an important and poorly understood topic related to environmental change in the region. However, two key

points that I think need to be clarified or addressed are topographic gradients and mortality. Microsite effects brought on by topographic variation seem to not be considered here, though they are an important consideration in a study measuring the rate and manner of treeline advancement. In addition, there is no discussion of mortality rates, both in seedlings and seeds. Seedling dispersal and seed/ling mortality are tightly interconnected and should be at least discussed if not reported. Overall however, this manuscript should be accepted with some of these modifications addressed.

Here are some other comments/questions/edits: • What makes LAVESI a spatially explicit model? It would be good if the authors could explain this in a few sentences. Even though the model has been previously published, it helps orient the reader to explain the model and what makes the model unique. • This parameterization as well as the improvements made to LAVESI concerning seed dispersal rates and distances were made based on data collected over a 100m x 100m plot. The size of this plot is quite small to base landscape scale conclusions on. The disadvantages of this plot size are not well discussed in the discussion. • What about topography? Topography is not mentioned and is a very important feature with respect to treeline advancement, seed dispersal rates/distances and seed viability. Microsite climate effects caused by topography are also not addressed. These too are very important to consider here. • The methods surrounding how the needle genotyping (2.2) was used to infer microsatellite data (2.3), and was then used to update seed disperal rates and distances in LAVESI are very confusing. It is unclear what was simulated and how, versus what was measured in the field. For ex., line 108, "We simulated the heritage for 10,000 seeds. . ." How was this simulated? With LAVESI? With a statistical model? With CERVUS 3.0.7? These sections are very confusingly written, readers would not be able to use them to reproduce your study. Please explain more clearly the steps that were taken to go from needle collection to LAVESI updates. • Lines 19-24: The writing is not clear whether the comparison was done before the model code updates or if the model was run on transects to address the shortcoming. • Section 2.4.1: The model though published elsewhere should be explained in a few more sentences

here. Why is it considered spatially explicit? What does that mean for this study in particular? How were the listed updates implemented? • Line 169: "Simulated" is more colloquial terminology than "hypothetical" • Lines 190, 195-196: Are these two different results? What is the difference between "pairs of larch individuals" and "two individuals within a clonal group"? • Line 237: Overemphasize is misspelled • Lines 322-332: Were these other study's all simulated or field-based results?

---

## Referee Comment (RC4) · Anonymous Referee #4 · 22 Nov 2018

This study utilizes genotyping and parentage analysis of individual trees to improve larch seed dispersal simulation within an individual-based, spatially-explicit forest model. The study is carried out at a single 100 m x 100 m site in the Taymyr Peninsula in northern Siberia. LAVESI, the forest model used, is specifically designed for individual larch growth, mortality, and regeneration, and the updated model is used to simulate northward migration of the larch treeline and forestline under two different climate scenarios. The updated model performed well when compared to observation data, though it slightly overestimated the number of recruits close to the parent tree as well as an overestimation of very long dispersal. The south-north migration simulation under static climate resulted in a migration rate of 0.6 m/year and 1.6 m/year for the

forest- and treelines, respectively. Under a climate scenario of decreasing temperature and slightly increasing temperature from south to north, the south-north migration rate was slower. They also found an accelerating rate of dispersal over the simulation time under the static climate scenario.

The study is important for field ecologists as well as the ecological modeling community. Currently, northward tree migration across the circumpolar boreal region is of crucial importance due to its potential impact and feedback to climate. However, most forest models do not adequately represent dispersal mechanisms. This study showcases an innovative way to determine in situ effective seed dispersal and incorporate such data into a forest model for calibration and application.

While the study is effective and well-structured, and shows how well the LAVESI model can perform at a local-scale, the model was tuned quite heavily to the small study area (only 100 m2), and the model output was compared only to data that was used in the tuning process. Before this model can be utilized at a larger scale I believe it will require more generalized parameter values. In particular, because the model produced fairly slow migration rates compared to other studies, I feel it may be overfitted to this study site and data, though only additional comparisons and simulations with the model will be able to determine if this is the case. It would be nice to see a sentence or two acknowledging this in the Conclusions. It would be nice in future studies to see this model compared to independent data at a separate site as well. I would also be interested to see how the migration would play out under a climate change scenario, though this is likely planned for future work.

Overall, I think this paper is well-written and the manuscript should be accepted with only a few minor revisions. This study is a great starting point for future work with this model and the equations developed within it. It should be of interest to other ecologists working on similar problems across the boreal region.

Below are some minor comments and edit suggestions for consideration by the authors:

Line 92: Change "Subsequent" to "Subsequently," Line 120: Change "larch species" to "larch individuals" Line 129: You say here and in the Supplementary Material that active layer depth influences tree mortality (which I am guessing is based on growth rate). However, it seems based on the information in the Supplementary Material that active layer depth directly influences tree growth, which in turn would also influence mortality (and potentially seed dispersal?). Lines 135-140: I'm not sure why some of these parameter descriptions are in quotes and some aren't. In general this sentence is difficult to get through. You may want to consider just publishing a table instead of listing them in the text. Line 139: I'm not sure what "different modes to compute the competition" are Line 151-152: Could you expand on the 20mx20m vs. the surrounding 100mx100m section? I'm not sure I follow where the spatial differences are coming from. Line 224: Add "for this model" after "Mean dispersal" Line 229: change "have the smallest" to "has the smallest"

Supplement S2: Line 74: Change "correspondingly" to "corresponding" and delete "roughly" Line 76: Change "of Matlack" to "from Matlack" Lines 76-79: I'm confused by what 0.86 m/s is referring to. I this Vd? Or w? Additionally this sentence is somewhat awkward and I would recommend breaking it up into two sentences and clarifying. Line 84: how did you obtain the sdist and the scaling parameter? I see that you tuned them variously but did you have initial starting values based on literature or data? Line 88: Where did you obtain the data for the study showing no significant influence of temperature? Was it at the same study site? I am concerned about this growth function modification as it further "tunes" the model to a specific area, and may need to be re-tuned if the model is moved elsewhere Lines 93-98: See my above comment on permafrost-tree growth influence. It seems ALT impacts tree growth directly and mortality indirectly, though I may be wrong. Line 97: What is the parameter fe? Table S4: I would suggest also adding variable symbols next to the parameter descriptions, especially if they are mentioned in this text or other published works. Line 105: Why

do you need to shift the dispersal peak by 2-3 m? Is this based on comparisons with the observation data? I would mention this here. Line 120: What is the reference simulation? Additionally please expand on what you mean by "general performance." Line 123: I'm not sure what you mean by "In parts" Line 127: What is the ecological basis for changing the density competition to improve the on-site recruitment ratio? Line 128: Delete "were" in between results and strongly

---

## Author Comment (AC1) · 19 Dec 2018

> **We thank the reviewer for the review and helpful comments. We revised our manuscript at the corresponding positions for each specific comment below.**
> **Our response are placed in bold font below each of the reviewer's comments in italics. Followed by a citation of changed text with a line statement that refers to the version of the manuscript with tracked changes.**

**General comments:**

*The manuscript by Kruse et al. describes an empirical study of effective seed dispersal using molecular markers which is used to adapt and parametrize a simulation model on larch migration rates at the arctic treeline. The topic of this study is of general interest because estimates for the capability of species to shift their distribution ranges are important for assessing the impact of climate change on many ecosystems. Especially tree species are of interest since they are the foundation species of many ecosystems and exhibit life history traits, which make direct observations difficult. The study offers a nice example for a combined approach with empirical data and simulations although its direct implications are somewhat limited due to the lack of replications and the rather small study site. However, the authors acknowledge these limitations in their manuscript and it will be nice to see some replicates in the future to judge the range of migration rates possible at different locations.*
*The manuscript is overall well written and clearly structured. The applied methods are well chosen and experiments and data analyses are described in sufficient detail. The results are discussed in a concise way using the available body of literature and conclusions are well founded.*
*In general the manuscript is of high quality and I have not found any major flaws. Please find below a few specific comments.*

**Specific comments:**

*Page 2 lines 53 ff: These sentences are a bit hard to follow. Also it does not immediately become clear that the authors refer only to the Taymyr peninsula. Since there is so much literature on this available, it should maybe added somewhere that this region is well studied, which is a further argument for chosing this region for the study.*

> **We added a reference to the region in focus here and tried to point out in the preceding sentence that this region was in focus for several treeline studies.**
>
> **Now the text in line 52ff is:**
> **"It represents an ideal study area because the treeline is formed of monospecific tree stands of *Larix* Mill. Taxa and was thus the focus of several treeline studies (IPCC, 2013; Naurzbaev et al., 2002; Sidorova et al., 2010). The response to warming seems to differ with time-scale:**

while millennial-scale warming during the mid-Holocene is reflected by a treeline location 200 km further north on the Taymyr Peninsula (Andreev et al., 2002; Klemm et al., 2016; MacDonald et al., 2008), the decadal-scale ongoing warming generates no response (Niemeyer et al., 2015; Wieczorek et al., 2017), possibly because of low seed availability."

*Page 3, lines 70 ff: I suggest to elaborate a bit more on the specific aims of the study here at the end of the introduction. Some aspects have been mentioned in earlier paragraphs but rather indirectly and not specifically related to this study.*

As suggested, we edited the final paragraph of our introduction to enhance the visibility of the specific aims of our study.

Lines 75ff are now:
"With this study, we aim at improving seed dispersal and establishment processes in the simulation model LAVESI to make it applicable for simulating treeline migration rates. Therefore, we undertook a genetic parentage analysis of a treeline stand on the southern Taymyr Peninsula by applying an assay of eight nuclear microsatellites to get a reliable estimate of the effective seed dispersal distance (1). This information was used to improve the individual-based model LAVESI (2), which we then ran to simulate treeline advances into the tundra and estimate migration rates (3)."

*Page 9, lines 195 f: Is there an explanation how ramet pairs can occur 30 m apart in Larix? How about the chance for full sibs to have identical genotypes?*

We carefully checked the possibility of having sampled the same tree again and are confident to not have such a sampling error in our data.

Indeed, it seems a rather far distance, however, further own observations support the existence of ramets even on a longer distance than under the crown.

1. Wind thrown trees can survive (own observations) and most probably produce adventitious roots from their branches when touching the ground (Kajimoto, 2010; Cooper, 1911). If the main stem rots, two separate individuals could be found if both survive.

2. Under non-favorable conditions, larches can survive forming krummholtz individuals (Wieczorek et al., 2017, own observations). They grow by producing horizontal branches rather than an upright stem. At these, they could form adventitious roots and again after separating two individuals could be found sharing the same genotype.

Nevertheless, there is a chance that full siblings share the same genotype, but which is quite low especially because we used highly polymorphic nuclear microsatellites that are not in linkage disequilibrium (Kruse et al., 2018). Thus, the probability that two full siblings would have the same given genotype under pure Mendel inheritance (without mutations and recombination) is approximately $\frac{1}{4}$ per locus. In our case this leads to a low chance of 1 to ~65536 ($\frac{1}{4}^8$) that full sibs share the same genotype.

Additionally, we cannot rule out that selfing or back-crossing have occurred that could yield to offspring being genetically identical to one

of the parents. If those modes of inheritance occur regularly and would have caused a misidentification of full siblings as clones, we would expect to observe an continuously increasing number of transitional states from identical genotypes (0 different alleles) to sharing 50% of their alleles (8 different alleles). However, it drops from the identified clonal groups to a very low value and increases again beginning at 3 to 4 differences (Fig. R1). This gives us confidence to classify such identical individuals as clones.

**Figure R1. For each individual the smallest number of different alleles, binned into 0 to a maximum of 16 alleles.**

[Figure]

References:
Cooper, W. S., 1911. Reproduction by layering among conifers. Botanical Gazette 52(5):pp. 369–379.
Kajimoto, T., 2010. Root system development of larch trees growing on siberian per- mafrost. In A. Osawa, O. A. Zyryanova, Y. Matsuura, T. Kajimoto & R. W.Wein, editors, Permafrost Ecosystems, volume 209, pages 303–330. Springer Nether- lands, Dordrecht.

*Page 15, line 320: When I read this, I asked myself if the model includes the case of established individuals ahead of the treeline, which are not able to reproduce, yet, because conditions do not allow this at the moment. When the conditions change, the treeline might progress quite rapidly at first and then slow down. Since I am not familiar with the model in detail, I cannot judge if this is a point worth discussing or a scenario worth simulating.*

For simplification and to clearly infer migration rates into tundra we did not allow in our transect simulation experiments survival of individuals ahead of the treeline until year 100.

At some places the presence of krummholtz may enhance migration if such tree island/refugials begin to reproduce sexually. Although not explicitly incorporated, this is partly covered by the homogenous forcing climate that allows long dispersed seedlings to survive with a higher chance ahead of the treeline than in the climate gradient scenario forming faster forest islands within the tundra (see simulation example

in Fig. 2). At the beginning of the simulations the migration rate is only slightly faster but this benefit accumulates over time until the positive effect can clearly be seen at the end of the simulation period (Fig. 6 & 7).

A detailed simulation study for a variety of latitudinal treelines might be worth considering in an extra study in which we could assess responses of different treeline types (e.g. sharp boundary vs. wide transition zone, and the presence/absence of krummholtz).

*Page 15, lines 325 ff: The migration rates mentioned here, are they 20-60 m/20-50 m for the entire time period respectively or per year in these time periods? Is it possible to translate the elevational shift into a migration rate comparable to the model?*

Migration rates at altitudinal treelines are hardly comparable to those of latitudinal treeline, the climate gradient is much steeper so that still seed sources are closer to the species limit so that climate improvements can lead to faster a migration response up the slope. In comparison, the same climate gradient is very likely thousand times longer on latitudes: 1 °C per ~150 m elevation compared to 1 °C per ~160.000 m latitude for the Taymyr Peninsula.

Because of not being strongly affected by dispersal limitations, they help to understand how treelines could ideally migrate when not limited by seed availability. Nevertheless, other restrictions might become more important for the migration process at these locations such as facilitation (e.g. Martínez et al., 2011).

References: Martínez, I., Wiegand, T., Camarero, J. J., Batllori, E. and Gutiérrez, E.: Disentangling the formation of contrasting tree-line physiognomies combining model selection and Bayesian parameterization for simulation models., Am. Nat., 177(5), E136–E152, doi:10.1086/659623, 2011.

*Page 15, line 330: Establishment will for sure be affected not only by density-dependent mortality but also by abiotic conditions and their stochasticity in this extreme region of the planet.*

That is right, therefore establishment of seeds dependent on weather forcing and their survival (mortality) implemented as a stochastic process (see details in Kruse et al., 2016).

**Technical comments:**

*Page 5, line 102: "inferred" Why are the microsatellite data described as inferred? To me they seem quite directly measured.*

The reviewer is right, the fragment lengths were measured and the parentage was inferred from these data. Accordingly, we deleted the word "inferred".

*Figure 5: The x axis is quite cramped in this figure. Maybe it could be stretched out a bit?*

For a better visibility of the plotted results, we show now lines instead of points and stretched the scale of the x-axis.

**The figure in line 236 is now:**

---

## Author Comment (AC2) · 19 Dec 2018

**We thank the reviewer for reviewing the manuscript and for the helpful comments. We revised our manuscript at the corresponding positions for each specific comment below.**
**Our response are placed in bold font below each of the reviewer's comments in italics. Followed by a citation of changed text with a line statement that refers to the version of the manuscript with tracked changes.**

**General comments:**

*This paper uses the LAVESI model to explore the rate of northward Larix migration on the southern portion of the Taimyr Peninsula in Siberia. The study uses recent field data collected in the area to drive the simulation work. The focus of the model parameterization/calibration was to improve the Larix dispersal functionality using genetic parentage, and then apply the model to understand how the Larix dispersal characteristics will play out in given some climate scenarios. The site of this modelling work is of particular interest, as it is near the northernmost forest stands where paleoecological records have indicated past presence of trees north of where they are currently found.*
*Given the unique nature of Larix forests (deciduous conifers linked to continuous permafrost distribution) and the fact that across their broad spatial domain (central and eastern Siberia) this class of trees coincides with marked changes in climate, and the potential for changes in tree distribution to alter the dynamics of high latitude systems (through changes to albedo and permafrost), this study is of great interest. I think the paper is for the most part clear and well-structured. Perhaps there can be some modification of the Discussion based on my main critique.*

**Main critiques:**

*1. At the scale of this study, are south-north assumptions of tree migration robust? At fine scales, the migrations may occur according to the patterns of favorable microsites, and primarily be confined to corridors with favorable active layer dynamics, and direct insolation. Such landforms seem important for explaining the current pattern of trees. A study with such an individual model that doesn't account for the conditions that are associated with the germination, survival, and growth of those individuals that are being dispersed should probably discuss in some detail this issue of the micro-site constraints that may contribute to broad error bars associated with migration rates. In other words, would the velocity of migration change between different microsites? If so, what is the relative prevalence of such favorable sites across the landscape, and how are they connected to the seed sources? These may be questions for follow-up work, and may be beyond the scope of this study, but I think a section in the Discussion could serve as a link between this study and some potentially viable next steps (one that incorporates landforms and micro-sites details).*

We added a discussion about the important microsite effects and seedling survival rates to section 4.3. "Treeline migration rates". Basically, we implicitly take account for them with our parameterization approach (Kruse et al., 2016). An explicit implementation would of course improve the realisticity of the model's outcome, but also increase the already high demand of parameters and finding good estimates for them. Nevertheless, testing for microsite effects and implementing them in the model would allow for a detailed study of their impact, but this is out of the scope of this manuscript.

Same response as to comment R3 General comments.

Line 361ff:
"Furthermore, the probability of seeds surviving and forming a seedbank and the survival rates of seedlings strongly determine the colonisation speed. This is linked to the availability of microsites where seedlings benefit from shelter, thus lowering their mortality rates (e.g. Resler et al., 2005; Maher et al., 2006; Germino et al., 2011). These effects are not explicitly simulated but implicitly taken account of by our model parameterisation (Kruse et al., 2016). Migration corridors along rivers are not taken into account but they likely assist colonisation in these landscapes because of deeper active-layer depths close to the rivers and also from downstream seed dispersal (Neilson et al., 2005; Wieczorek et al., 2017). Nevertheless, the positive impact of an increased survivorship on migration rates can be observed in our migration simulation experiments.

The mortality rate ahead of the treeline is lower under homogeneous climate than in the linearly decreasing climate gradient scenario with the consequence that the migration enters the exponential phase earlier (Fig. 6 & 7). In addition, we based our model adaptations on an area that is only one hectare in size and with this we cannot directly assess the long-distance seed dispersal to which to fit our implemented kernel. To account for these cases, we implemented a Gaussian dispersal kernel combined with an exponential shaped with a fat tail (Kruse et al., 2016). In this study, this allows numerous seeds to be dispersed to far distances and led to a higher immigration into the simulated forest plot than observed. In consequence, the simulated migration rate tends to be overestimated.

This comprehensive study from genetic analyses to a model application is a first attempt showing the importance of undertaking these timely model parameterisation studies and should be enhanced by, for example, inferring the parentages for other positions in the treeline ecotone on the southern Taymyr Peninsula."

*2. This area is the northernmost forest ecotone. Some discussion for why this may be the case (paleoecological history) could be interesting and help contextualize predicttions of future treeline velocities.*

We added a short history of the treeline at the Taymyr Peninsula and giving likely explanations for the northward expansion and rate, ending with the modern situation.

**In lines 350ff:**

**"During the Holocene Thermal Maximum boreal forests expanded on the Taymyr Peninsula to their northernmost position during the Holocene, which was likely assisted by glacial refugial populations ahead of the treeline (MacDonald et al., 2000, 2008). The treeline responded with a centennial lag to environmental improvement, for example solar insolation, and reached its maximum position at ~8000 to 4000 yr BP, and subsequently declined to reach its modern limits around 3000 yr BP (MacDonald et al., 2000). Recently, global warming is ameliorating conditions for _Larix_ forests in Siberia and evidence can be found that treeline stands are starting to respond, but at a slower rate than one might expect given the strong increase in temperatures (Wieczorek et al., 2017; Harsch et al., 2009)."**

_3. In Figure 1, a simple schematic of the sampling design could be useful._

**We added a simple sampling scheme of trees for the study site in Fig. 1.**

**The figure now is in line 93:**

[Figure]

**Other comments:**

*In the abstract, might it be possible to replace some of the technical wording associate with the genetic analysis with other more recognizable terminology that would be more likely to be understood by most of the readers of this journal?*

**This comment refers to the sentence in line 17. Following the suggestion of the reviewer, we edited this by deleting "highly polymorphic", which is already a requirement for the parentage analysis that we meet with our eight microsatellite loci. Additionally, we exchanged the word "loci" and used the synonym "marker".**

**The text in line 16ff is now:**
**"We inferred the effective seed dispersal distances of a typical open forest stand on the southern Taymyr Peninsula (north-central Siberia) from genetic parentage analysis using eight nuclear microsatellite markers."**

*Figure 4. Nice figure that shows how a few individuals dominate the reproduction. I think it needs to be graphically enhanced. Suggestion: Put an alpha (i.e., transparency) on the green lines, and put the dots on top.*

**As suggested we changed the transparency of the dispersal connections. Furthermore, we use now lines instead of arrows for a clearer view on the connections. Bringing the dots to the front caused many connections to be hidden. So that we decided to modify the coordinate system by stretched it to zoom in to the plot centre where most recruits and their connection to the parents can now be seen.**

[Figure]

*Section 4.3 Line 336 - "However, further processes...and should therefore [NOT?] be neglected in simulation studies:..."*

**Yes, this is a critical point for us. We corrected this mistake in writing.**

---

## Author Comment (AC3) · 19 Dec 2018

**We thank the reviewer for reviewing our manuscript and for the helpful comments. We revised our manuscript at the corresponding positions for each specific comment below.**
**Our response are placed in bold font below each of the reviewer's comments in italics. Followed by a citation of changed text with a line statement that refers to the version of the manuscript with tracked changes.**

**General comments:**

*This study presents seed dispersal improvements made to the LAVESI individual-based, spatially explicit vegetation simulator to investigate larch migration rates on the Taymyr Peninsula in Northern Siberia. The seed dispersal equation updates were based on a field-based study of genetic parentage of trees in the study area, which found that prior to improvement, the model tended to underestimate dispersal distances in general and overestimate the numbers of recruits close to the parent tree.*
*The updated model was used to simulate south-to-north transects and the rate of tree line advance was found to differ from the rate of forest line advance by ~1m per year. The study is well-structured and presented, and it addresses an important and poorly understood topic related to environmental change in the region.*

*However, two key points that I think need to be clarified or addressed are topographic gradients and mortality. Microsite effects brought on by topographic variation seem to not be considered here, though they are an important consideration in a study measuring the rate and manner of treeline advancement. In addition, there is no discussion of mortality rates, both in seedlings and seeds. Seedling dispersal and seed/ling mortality are tightly interconnected and should be at least discussed if not reported. Overall however, this manuscript should be accepted with some of these modifications addressed.*

**We added a discussion about the important microsite effects and seedling survival rates to section 4.3. "Treeline migration rates". Basically, we implicitly take account for them with our parameterization approach (Kruse et al., 2016). An explicit implementation would of course improve the realisticity of the model's outcome, but also increase the already high demand of parameters and finding good estimates for them. Nevertheless, testing for microsite effects and implementing them in the model would allow for a detailed study of their impact, but this is out of the scope of this manuscript.**

**Same response as to comment R2 1.**

Line 361ff:
"Furthermore, the probability of seeds surviving and forming a seedbank and the survival rates of seedlings strongly determine the colonisation speed. This is linked to the availability of microsites where seedlings benefit from shelter, thus lowering their mortality rates (e.g. Resler et al., 2005; Maher et al., 2006; Germino et al., 2011). These effects are not explicitly simulated but implicitly taken account of by our model parameterisation (Kruse et al., 2016). Migration corridors along rivers are not taken into account but they likely assist colonisation in these landscapes because of deeper active-layer depths close to the rivers and also from downstream seed dispersal (Neilson et al., 2005; Wieczorek et al., 2017). Nevertheless, the positive impact of an increased survivorship on migration rates can be observed in our migration simulation experiments.

The mortality rate ahead of the treeline is lower under homogeneous climate than in the linearly decreasing climate gradient scenario with the consequence that the migration enters the exponential phase earlier (Fig. 6 & 7). In addition, we based our model adaptations on an area that is only one hectare in size and with this we cannot directly assess the long-distance seed dispersal to which to fit our implemented kernel. To account for these cases, we implemented a Gaussian dispersal kernel combined with an exponential shaped with a fat tail (Kruse et al., 2016). In this study, this allows numerous seeds to be dispersed to far distances and led to a higher immigration into the simulated forest plot than observed. In consequence, the simulated migration rate tends to be overestimated.

This comprehensive study from genetic analyses to a model application is a first attempt showing the importance of undertaking these timely model parameterisation studies and should be enhanced by, for example, inferring the parentages for other positions in the treeline ecotone on the southern Taymyr Peninsula."

Here are some other comments/questions/edits:

*What makes LAVESI a spatially* explicit model? It would be good if the authors could explain this in a few sentences. Even though the model has been previously published, it helps orient the reader to explain the model and what makes the model unique.

We added a short descriptions what our model makes it an individual-based model and explained the advantages of such a detailed approach.

The edited text can be found in line 58ff:
"To study the responses and migration dynamics of treeline tree stands under climate change, LAVESI, an individual-based and spatially explicit simulation model for *Larix* (Kruse et al., 2016; Wieczorek et al., 2017), was developed. In comparison to other dynamic vegetation models, it handles each individual larch tree beginning from a seed to an established seedling until becoming a mature tree and producing seeds itself and thus starting a new generation. This model includes wind-

**dependent seed dispersal and density-dependent growth and mortality processes. The representation of the full life cycle allows in-detail simulation experiments to unravel the influences of previously overlooked feedbacks (further details in Kruse et al., 2016; Wieczorek et al., 2017)."**

*This parameterization as well as the improvements made to LAVESI concerning seed dispersal rates and distances were made based on data collected over a 100m x 100m plot. The size of this plot is quite small to base landscape scale conclusions on. The disadvantages of this plot size are not well discussed in the discussion.*

**We extended the discussion about the plot size of 100x100 m. This area is at the upper edge to be manageable during expeditions to these remote areas. Several people needed days to record and sample these <1000 individuals. However, at more densely populated forests plots we sampled >3000 individuals at similar areas or even on smaller plots.**

**We decided to use the northernmost plot close to the species line as this is the likely area responding most strongly and very likely "preparing" for a northwards migration triggered by recent climate warming.**

**A larger area does mean more work and we think that the knowledge gain does not scale with effort. We added here sentences and also under 4.3 first paragraph at the end.**

**In line 284ff:**
**"Unfortunately, the labour-intensive sample collection and genetic analyses restricted the analysis to a rather small area in comparison to the large area of the treeline transition zone. Assessing the parentage across a broader scale and for different positions in the treeline ecotone would further help to understand dispersal dynamics at the treeline but the additional knowledge gain does not scale with effort."**
**In line 378ff:**
**"This comprehensive study from genetic analyses to a model application is a first attempt showing the importance of undertaking these timely model parameterisation studies and should be enhanced by, for example, inferring the parentages for other positions in the treeline ecotone on the southern Taymyr Peninsula."**

*What about topography? Topography is not mentioned and is a very important feature with respect to treeline advancement, seed dispersal rates/distances and seed viability. Microsite climate effects caused by topography are also not addressed. These too are very important to consider here.*

**See our response to the general comment above.**

*The methods surrounding how the needle genotyping (2.2) was used to infer microsatellite data (2.3), and was then used to update seed disperal rates and distances in LAVESI are very confusing. It is unclear what was simulated and how, versus what was measured in the field. For ex., line 108, "We simulated the heritage for 10,000 seeds..." How was this simulated? With LAVESI? With a statistical model? With CERVUS 3.0.7? These sections are very confusingly written, readers would not be able to use them to reproduce your study. Please explain more clearly the steps that were taken to go from needle collection to LAVESI updates.*

We checked the sequence of the regarding methods and edited section 2.3. to made more clear that the observed parentages were estimated in the program CERVUS and not with our model LAVESI. Following the first method sections about the field data and subsequent analysis until estimating effective seed dispersal distances, we introduce the model tuning steps in section 2.4.2.

Line 115ff:
"We determined parents from allele frequency data with a likelihood-based approach implemented in CERVUS version 3.0.7 (Kalinowski et al., 2007). During the analyses, we allowed for 1% of errors in genotyping and a minimum of seven loci typed in the final analysis. All individuals (612 in total) were analysed and we searched for parents of recruits (height <2 m) from among all potential tree individuals (height >0.4 m). Following the program documentation we simulated in CERVUS the heritage for 10,000 seeds with a chance of 10% of a parent sampled and 1% error (Marshall et al., 1998; Slate et al., 2000) to determine thresholds for the 'log of the overall likelihood ratio' (LOD) scores in this analysis."

*Lines 19-24: The writing is not clear whether the comparison was done before the model code updates or if the model was run on transects to address the shortcoming.*

We clarified which model version we used for the transect simulation.

Now text in line 21f:
"We thus adapted our model and used the newly parameterised version to simulate south-to-north transects: a slow-moving treeline front was revealed."

*Section 2.4.1: The model though published elsewhere should be explained in a few more sentences here. Why is it considered spatially explicit? What does that mean for this study in particular? How were the listed updates implemented?*

We added the requested details in the introduction. In addition, we edited the methods section 2.4.2, but we refer the reader to the supplement 2 for the technical description of the model tuning by modifying parameters or newly introduced variables.

Line 58ff:
"In comparison to other dynamic vegetation models, it handles each individual larch tree beginning from a seed to an established seedling until becoming a mature tree and producing seeds itself and thus starting a new generation. This model includes wind-dependent seed dispersal and density-dependent growth and mortality processes. The representation of the full life cycle allows in-detail simulation experiments to unravel the influences of previously overlooked feedbacks (further details in Kruse et al., 2016; Wieczorek et al., 2017)."

Line 145ff:
"To tune the model's processes in order to capture the observed effective seed dispersal distribution, we tested several combinations of model parameters and introduced new variables into formulae used in the program code of the model (listed in Table 1, details in Supplement 2 and in Table S5)."

*Line 169: "Simulated" is more colloquial terminology than "hypothetical"*

**Done, changed to "simulated"**

*Lines 190, 195-196: Are these two different results? What is the difference between "pairs of larch individuals" and "two individuals within a clonal group"?*

**The 11 individuals are those that are the excluded individuals from further analyses, which were part of the 10 clonal groups consisting of 22 individuals (9x2 and 1x4 individuals). We edited the text for clarification.**

**Line 210ff:**
**"In total, 601 sampled trees could be distinguished and 22 individuals were identified as 10 clonal groups, of which 11 were subsequently excluded from further analyses (Fig. 3a, Supplement S1). The maximum distance between two individuals within these groups was 30 m but mostly <5 m (Fig. 3a)."**

*Line 237: Overemphasize is misspelled*

**Done**

*Lines 322-332: Were these other study's all simulated or field-based results?*

**They were all field-based studies and we added a reference to that in the sentence for clarification.**

**Line 346ff start now with:**
**"Another field-based study reports […]"**

---

## Author Comment (AC4) · 19 Dec 2018

**We thank the reviewer for reviewing our manuscript and especially for a closer look on the supplementary material. The comments helped to improve the first version of our manuscript. This was revised at the corresponding positions for each specific comment below.**
**Our response are placed in bold font below each of the reviewer's comments in italics. Followed by a citation of changed text with a line statement that refers to the version of the manuscript with tracked changes.**

**General comments:**

*This study utilizes genotyping and parentage analysis of individual trees to improve larch seed dispersal simulation within an individual-based, spatially-explicit forest model. The study is carried out at a single 100 m x 100 m site in the Taymyr Peninsula in northern Siberia. LAVESI, the forest model used, is specifically designed for individual larch growth, mortality, and regeneration, and the updated model is used to simulate northward migration of the larch treeline and forestline under two different climate scenarios. The updated model performed well when compared to observation data, though it slightly overestimated the number of recruits close to the parent tree as well as an overestimation of very long dispersal. The south-north migration simulation under static climate resulted in a migration rate of 0.6 m/year and 1.6 m/year for the forest- and treelines, respectively. Under a climate scenario of decreasing temperature and slightly increasing temperature from south to north, the south-north migration rate was slower. They also found an accelerating rate of dispersal over the simulation time under the static climate scenario.*
*The study is important for field ecologists as well as the ecological modeling community. Currently, northward tree migration across the circumpolar boreal region is of crucial importance due to its potential impact and feedback to climate. However, most forest models do not adequately represent dispersal mechanisms. This study showcases an innovative way to determine in situ effective seed dispersal and incorporate such data into a forest model for calibration and application.*

*While the study is effective and well-structured, and shows how well the LAVESI model can perform at a local-scale, the model was tuned quite heavily to the small study area (only 100 m2), and the model output was compared only to data that was used in the tuning process. Before this model can be utilized at a larger scale I believe it will require more generalized parameter values. In particular, because the model produced fairly slow migration rates compared to other studies, I feel it may be overfitted to this study site and data, though only additional comparisons and simulations with the model will be able to determine if this is the case. It would be nice to see a sentence or two acknowledging this in the Conclusions. It would be nice in future studies to see this model compared to independent data at a separate site as*

*well. I would also be interested to see how the migration would play out under a climate change scenario, though this is likely planned for future work.*

*Overall, I think this paper is well-written and the manuscript should be accepted with only a few minor revisions. This study is a great starting point for future work with this model and the equations developed within it. It should be of interest to other ecologists working on similar problems across the boreal region.*

> **Response to the the centre part of the general comment in starting with "While […]". A similar comment came from R3. We added a short discussion about the "small" study area that is already challenging for such an analysis to the discussion in section 4.1. Nevertheless, it would be worth to undergo this work at more sites to compare the findings of this study to other treeline locations.**
>
> **Line 284ff:**
> **"Unfortunately, the labour-intensive sample collection and genetic analyses restricted the analysis to a rather small area in comparison to the large area of the treeline transition zone. Assessing the parentage across a broader scale and for different positions in the treeline ecotone would further help to understand dispersal dynamics at the treeline but the additional knowledge gain does not scale with effort."**
>
> **Additionally, we extended our conclusion covering the comment on further studies that would help unravelling if our slow migration rate estimate is flawed by overfitting to only one study site or not, as requested by the reviewer.**
>
> **Line 400ff:**
> **"To find out if the estimated slow migration is an outlier coming from overfitting to only one study site or the general response rate under current warming, further similar studies at other treeline positions would be necessary."**

**Below are some minor comments and edit suggestions for consideration by the authors:**

*Line 92: Change "Subsequent" to "Subsequently,"*

> **Response: Done**

*Line 120: Change "larch species" to "larch individuals"*

> **Response: Done**

*Line 129: You say here and in the Supplementary Material that active layer depth influences tree mortality (which I am guessing is based on growth rate). However, it seems based on the information in the Supplementary Material that active layer depth directly influences tree growth, which in turn would also influence mortality (and potentially seed dispersal?).*

> **We use the actual tree growth in comparison to the maximum potential growth of the same tree as currency for productivity and mortality.**
>
> **The given information was not sufficient to explain how active layer depth influences trees (growth/mortality). In consequence, we edited the text in the Methods section for clarification.**

**Line 137ff:**
**"The original model of Kruse *et al.* (2016) was updated with the following processes (details in Suppelement 2): (i) seed dispersal distances now depend on species-specific traits (tree height, seed properties) and wind speed and direction (Kruse et al., 2018b), (ii) the tree diameter growth function is newly calibrated to the climate forcing (Epp et al., 2018), and (iii) the active-layer thaw depth directly influences the tree's growth that is used to estimate it's seed production and mortality."**

*Lines 135-140: I'm not sure why some of these parameter descriptions are in quotes and some aren't. In general this sentence is difficult to get through. You may want to consider just publishing a table instead of listing them in the text.*

**For clarification we decided to remove the listing of only some of the varied parameters and refer readers to the complete information in the supplementary material. A complete list and further detailed information on each parameter combination and the process can be found there.**

*Line 139: I'm not sure what "different modes to compute the competition" are*

**We tested the impact of several implementations of influence areas and strengths of competition on the trees diameter growth. The actual growth of an individual is the currency in the model by which other functionalities are based on (seed production/mortality).**

**We refer now the reader to the supplementary material, as there is the information on modified parameters/modes and tested model variables.**

*Line 151-152: Could you expand on the 20mx20m vs. the surrounding 100mx100m section? I'm not sure I follow where the spatial differences are coming from.*

**We needed to make the simulated data comparable to the inferred effective seed dispersal distances. Therefore, we followed directly our sampling scheme as described in Section 2.1 sample collection "[…] We sampled all individuals >0.4 m in height in a 20 x 20 m area as well as all trees >2 m high or bearing cones from the surrounding 100 x 100 m area (Fig. 3). Additionally, in the central 12 x 12 m area individuals <0.4 m were collected."**

**Here we added a reference to the sampling scheme description in section 2.1 in the regarding sentence.**

**Line 166ff:**
**"We resampled these simulated distances to consider the same frequency of observed parenthoods in the central 20 x 20 m as in the surrounding 100 x 100 m area (sampling scheme details in section 2.1 sample collection)."**

*Line 224: Add "for this model" after "Mean dispersal" Line 229: change "have the smallest" to "has the smallest"*

**Response: Done**

**Supplement S2:**

*Line 74: Change "correspondingly" to "corresponding" and delete "roughly"*

**Response: Done**

*Line 76: Change "of Matlack" to "from Matlack"*

**Response: Done**

*Lines 76-79: I'm confused by what 0.86 m/s is referring to. I this Vd? Or w? Additionally this sentence is some- what awkward and I would recommend breaking it up into two sentences and clarifying.*

**The value 0.86 m/s is referring to the descent rate for seeds, which is abbreviated by Vd. We separated the sentences as suggested and edited it for clarification.**

**The corrected part of the text is now in line 74ff:**
**"The release height $H_t$ is estimated at 75% of the individual's height. $V_d$ is the descent rate for seeds and is estimated for *Larix gmelinii* by a linear regression using species data from Matlack (1987). For species having wing-scales attached to the seeds, this rate can be calculated by $V_d = 0.0032 * \sqrt{w} + 0.4807$ and is 0.86 m s$^{-1}$, with the wing loading $w$ (Matlack 1987) for *L. gmelinii*. The variable $w$ is calculated by dividing the average seed weight (in microdyne) of 3.5 mg (Heit and Eliason, 1940; Lukkarinen et al., 2009) by the propagule area of 0.2 cm$^2$ (Fu et al., 1999)."**

*Line 84: how did you obtain the sdist and the scaling parameter? I see that you tuned them variously but did you have initial starting values based on literature or data?*

**When implementing the seed dispersal kernel into the model (Kruse et al., 2016), we made a first guess for the resulting dispersal kernel based on literature values and tuned those values to observed patterns.**

*Line 88: Where did you obtain the data for the study showing no significant influence of temperature? Was it at the same study site? I am concerned about this growth function modification as it further "tunes" the model to a specific area, and may need to be re-tuned if the model is moved elsewhere*

**We used a tree ring series from Yamal of the National Climatic Data Center data bank for *Larix sibirica* and own data for *Larix gmelinii* from Khatanga near the study site and for both data from the nearest weather station. For further information, please see the supplement of Epp et al. (2018) published in Scientific Reports.**

**Regarding the second part of the comment. The modelled tree diameter growth in the current version of the model is adapted to weather in Taymyr and Yamal. Therefore, it has to be tuned for each species and region when using it for further applications.**

*Lines 93-98: See my above comment on permafrost-tree growth influence. It seems ALT impacts tree growth directly and mortality indirectly, though I may be wrong.*

**Yes, answered in the other comment above.**

*Line 97: What is the parameter fe?*

**It is a soil property parameter, see definition in Hinkel and Nicholas (1995).**

*Table S4: I would suggest also adding variable symbols next to the parameter descriptions, especially if they are mentioned in this text or other published works.*

**We added for the model parameters the corresponding symbols. Corresponding changes were made in Table S5.**

*Line 105: Why do you need to shift the dispersal peak by 2-3 m? Is this based on comparisons with the observation data? I would mention this here.*

**We tried to explore potential setting to align the modelled effective dispersal distances to the observations. For clarification, we edited the sentence and refer to the results presented in the main article.**

**The text now line 105 reads:**
**"To fit the simulated seed effective dispersal distance to observations (Fig. 5) we explored potential settings …"**

*Line 120: What is the reference simulation? Additionally please expand on what you mean by "general performance."*

**We extended the statement of the reference simulation, which is the baseline simulation with the original model. Furthermore, we added for clarification of the "general performance" a reference to the correlation coefficients in Table S5.**

**This sentences in line 122ff changed to:**
**"This was improved by other simulations (qt-wJ) but their general performance (lower correlation coefficients, Table S5) was weaker than the reference simulation without parameter changes or adaptations of the model (a)."**

*Line 123: I'm not sure what you mean by "In parts"*

**We deleted the confusing beginning of the first sentence of the regarding paragraph. In the following sentence we briefly state the achievements, but also at which results the best fitting model version deviated from the observed pattern.**

**Text now in line 126:**
**"We achieved a good fit when increasing the peak of the dispersal function in the model to longer distances."**

*Line 127: What is the ecological basis for changing the density competition to improve the on-site recruitment ratio?*

**Similar to Janzen and Connell's findings, recruits have the highest chance to survive at intermediate distance to the producing tree, not directly at it. They are "pushed back" by the mother tree for a variety of reasons (shadow of the tree's crown, high pest pressure/seed predators,**

**exhausted nutrients in the active layer, insulating accumulation of needles and other litter, etc.).**

**This is implicitly implemented in the model and can be manipulated by varying the competition density, e.g. by increasing the influence on smaller trees. With this, seedlings from farther distances could have a likely higher chance to establish.**

*Line 128: Delete "were" in between results and strongly*

**Done**